# Natural image statistics for mouse vision

**Luca Abballe[1], Hiroki Asari[2]\***

**1** Department of Biomedical Engineering, Sapienza University of Rome, Rome, Italy, **2** European Molecular Biology Laboratory, Epigenetics and Neurobiology Unit, EMBL Rome, Monterotondo, Rome, Italy

\* asari@embl.it

## Abstract

The mouse has dichromatic color vision based on two different types of opsins: short (S)- and middle (M)-wavelength-sensitive opsins with peak sensitivity to ultraviolet (UV; 360 nm) and green light (508 nm), respectively. In the mouse retina, cone photoreceptors that predominantly express the S-opsin are more sensitive to contrasts and denser towards the ventral retina, preferentially sampling the upper part of the visual field. In contrast, the expression of the M-opsin gradually increases towards the dorsal retina that encodes the lower visual field. Such a distinctive retinal organization is assumed to arise from a selective pressure in evolution to efficiently encode the natural scenes. However, natural image statistics of UV light remain largely unexplored. Here we developed a multi-spectral camera to acquire high-quality UV and green images of the same natural scenes, and examined the optimality of the mouse retina to the image statistics. We found that the local contrast and the spatial correlation were both higher in UV than in green for images above the horizon, but lower in UV than in green for those below the horizon. This suggests that the dorsoventral functional division of the mouse retina is not optimal for maximizing the bandwidth of information transmission. Factors besides the coding efficiency, such as visual behavioral requirements, will thus need to be considered to fully explain the characteristic organization of the mouse retina.

## Introduction

Sensory systems have been considered to be adapted to the statistical properties of the environment through evolution [1]. Animals encounter different types of sensory signals depending on their natural habitats and lifestyles, and this can serve as an evolutionary driving force for each species to optimize its sensory systems for processing those signals that appear more frequently and are relevant for survival [2]. The optimality of the sensory processing has been broadly supported from an information theoretic viewpoint of coding efficiency [3,4]. In particular, various physiological properties of sensory neurons can be successfully derived from learning efficient codes of natural images or natural sounds, such as separation of retinal outputs into ON and OFF channels [5], Gabor-like receptive fields of visual cortical neurons [6], and cochlear filter banks [7]. Such computational theories and statistical models are, however, often limited to generic features of the sensory processing, and fail to account for species-specific fine details partly due to a lack of proper data sets of natural sensory signals.

**Data Availability Statement:** All relevant data and code underlying this study are publicly available at Zenodo with DOI 10.5281/zenodo.5204507.

**Funding:** This work was supported by research grants from EMBL (H.A.). The funder had no role in

study design, data collection and analysis, decision to publish, or preparation of the manuscript.

**Competing interests:** The authors have declared that no competing interests exist.

In the past decade, the mouse has become a dominant model for studying the visual system mainly because of the wide availability of experimental tools [8]. Compared to other mammalian model animals such as cats and primates, however, the mouse vision has certain distinctive properties. For example, mice are dichromats as many other mammals are, but their retina expresses ultraviolet (UV)-sensitive short (S)-wavelength sensitive opsins and green-sensitive middle (M)-wavelength sensitive opsins [9–11]. While UV vision is common in amphibians, birds and insects, it has not been identified in mammals except for a few species including rodents [12–14]. Moreover, the mouse retina has no fovea but a prominent dorsoventral gradient in the expression pattern of the two opsins [10,15–17]. A vast majority of the mouse cone photoreceptors ($\sim$95%) co-express the two opsins but with a dominant expression of S- and M-opsins in the ventral and dorsal parts of the retina, respectively [9,10,18,19]. This makes the upper visual field more sensitive to UV than green, and vice versa for the lower visual field [20]. It is natural to assume that this functional segregation of the mouse vision has evolved due to an adaptation to the natural light distribution as the sunlight is the major source of UV radiation. It remains unclear, though, how optimal the mouse visual system is to natural scene statistics *per se*.

While natural image statistics have been extensively studied thus far [1,21], those outside the spectral domain of human vision remain to be fully explored [2,18,22–24]. Here we thus developed a multi-spectral camera system to sample high-quality images that spectrally match the mouse photopic vision, and analyzed the statistics of the UV and green image data sets to test the optimality of the sampling bias in the mouse retina along the dorsoventral axis [9,10,18,19]. We identified distinct statistical properties in the UV and green channels between the upper and lower visual field images; however, these image statistics were not necessarily consistent with what the efficient coding hypothesis would predict from the functional organization of the mouse retina.

## Materials and methods

All data and codes are available on Zenodo (10.5281/zenodo.5204507).

### Multi-spectral camera

**Design.** We built a multi-spectral camera system based on a beam-splitting strategy [25,26] to acquire images of the same scenes with ultraviolet (UV)- and green-transmitting channels that match the spectral sensitivity of the mouse photopic vision (Fig 1A) [9–11]. The light coming from a commercial camera lens (Nikon, AF Nikkor 50 mm f/1.8D) was collimated with a near-UV achromatic lens (effective focal length, 50 mm; Edmund Optics, 65–976) and split with a dichroic filter (409 nm; Edmund Optics, 34–725). The reflected light, on the one hand, passed through a UV-selective filter set (HOYA U-340 and short-pass filter at 550 nm; Edmund Optics, 84–708) and formed the UV images focused on the first global-shutter camera (Imaging Source, DMK23UX174) with a near-UV achromatic lens (effective focal length, 50 mm; Edmund Optics, 65–976). The transmitted light, on the other hand, passed through a band-pass filter (500±40 nm; Edmund Optics, 65–743) and a lens (Edmund Optics, 65–976), and formed the green images sampled by the second camera (Imaging Source, DMK23UX174). To maximize the dynamic range of the two camera sensors (used with the same settings), we attenuated the light intensity of the green channel using an absorptive neutral density (ND) filter (optical density: 1.0, 1.3, 1.5, 1.8, or 2.0) on a filter wheel (Thorlabs, LTFW6) because the sunlight has much higher power in green than in UV (Fig 1B). The optical components are all mounted with standard light-tight optomechanical components (Thorlabs, 1-inch diameter lens tubes).

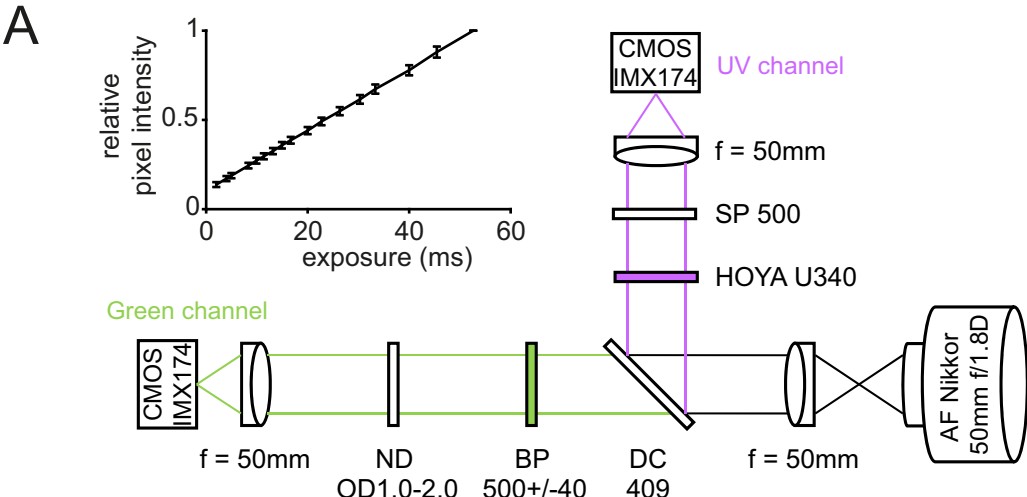

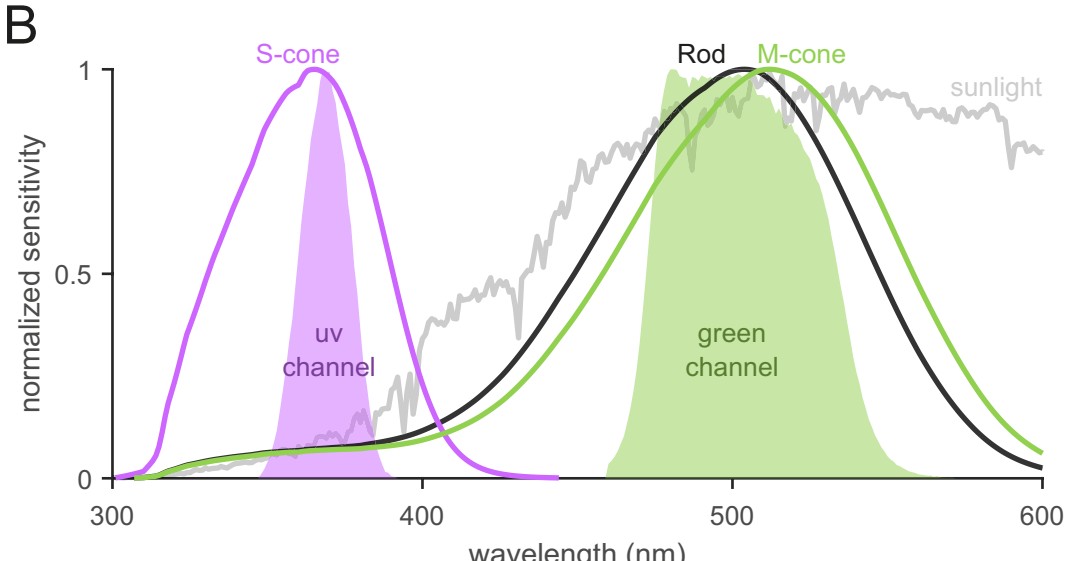

**Fig 1. Multi-spectral camera system for the mouse vision.** (A) Schematic diagram of the camera optics. Incoming light was split into UV and Green channels by a dichroic mirror and further filtered to match the spectral sensitivity of the mouse visual system (see panel B). A neutral density filter with the optical density value from 1.0 to 2.0 was used for the Green channel to maximize the dynamic range of the camera sensor to be used with the same parameter settings as the UV channel. The inset shows the pixel intensity values as a function of the exposure time (mean ± standard deviation; N = 2,304,000 pixels), supporting the linearity of the camera sensor (Sony, IMX174 CMOS). (B) Relative spectral sensitivity of the camera system (UV channel, violet area; Green channel, green area). For comparison, the spectral sensitivity of the mouse rod and S- and M-cone photoreceptors [31] corrected with the transmission spectrum of the mouse eye optics [30] was shown in black, violet and green lines, respectively, as well as typical sunlight spectrum in gray.

A recent study employed a similar design but with a fisheye lens to study the "mouse-view" images [22]. Our design has the following advantages over a panoramic camera design [22–24] to sample high quality image patches suitable for image statistics analysis. First, we chose a small field of view (11.3 degrees horizontally and 7.3 degrees vertically; 0.006 degrees/pixel) to minimize image distortion, and a large field of depth (the smallest aperture size on the Nikon

lens, *f*/22) to maximize areas in focus. This also allowed us to adjust camera settings (exposure length) to fully capture the dynamic range of individual scenes. Second, we chose a high-performance camera sensor (Sony, IMX174 complementary metal-oxide-semiconductor; CMOS) that has high quantum efficiency (~30% at 365 nm; ~75% at 510 nm), high dynamic range (73 dB; 12 bit depth), high pixel resolution (1920-by-1200 pixels), and linear response dynamics (Fig 1A, inset) [27–29].

**Spectral analysis.** The spectral sensitivity of the multi-spectral camera system (Fig 1B) was calculated by convolving the relative transmission spectra of the optics for each channel with the spectral sensitivity of the camera sensor (Sony, IMX174 CMOS) [29]. The relative transmission spectra were measured with a spectrometer (Thorlabs, CCS200/M; 200–1000 nm range) by taking the ratio of the spectra of a clear sunny sky (indirect sunlight) with and without passing through the camera optics.

For a comparison, we modelled the spectral sensitivity of the mouse visual system by convolving the transmission spectra of the mouse eye [30] with the absorption spectra of the mouse cone photoreceptors (Fig 1B). We used a visual pigment template [31] with the center frequency at 360 nm and 508 nm to simulate the short (S)- and middle (M)-wavelength-sensitive opsins in the mouse retina, respectively [9–11].

## Image acquisition

In total, we collected 232 images of natural scenes without any artificial object in the suburbs of Lazio/Abruzzo regions in Italy from July 2020 to May 2021. All the images were acquired using a custom-code in Matlab (Image Acquisition Toolbox) without any image correction, such as gain, contrast, or gamma adjustment. The two cameras were set with the same parameter values adjusted to each scene, such as the exposure length, and a proper ND filter was chosen for the green channel so that virtually all the pixels were within the dynamic range of the camera sensors (see examples in S2 Fig). Thus, our image data sets have no underexposed pixels and only a negligible number of overexposed pixels (0.0011% of pixels in 2 UV images and 0.0007% of pixels in 6 Green images). This is critical because the presence of under- or overexposed pixels will skew the image statistics.

When acquiring images, the camera system was placed on the ground to follow the viewpoint of mice. The following meta-data were also recorded upon image acquisition: date, time, optical density of ND filter in the green channel, weather condition (sunny; cloudy), distance to target object (short, within a few meters; medium, within tens of meters; or long), presence/absence of specific objects (animals; plants; water), and camera elevation angle (looking up; horizontal; looking down). We also took a uniform image of a clear sunny sky (indirect sunlight) as a reference image for vignetting correction (see below Eq (1)).

All the images were taken under ample natural light during the day. Although we did not measure the exact illuminance $\Phi$ of the environment, we expect that the lighting condition was on the order of $10^3 \sim 10^5$ lux (i.e., $\Phi = 10^7 \sim 10^9$ photons/$\mu$m$^2$/s). Assuming the mouse pupil diameter $d_{pupil} = 0.5$ mm, the eye diameter $d_{eye} = 4$ mm, the transmittance of the eye optics $T = 0.5$, and the light collection area of a photoreceptor $A_{photoreceptor} = 0.5$ $\mu$m$^2$, the photon flux on individual photoreceptors can then be estimated as $\Phi \cdot A_{pupil}/A_{retina} \cdot T \cdot A_{photoreceptor} = 10^4 \sim 10^6$ photons/photoreceptor/s, where $A_{pupil} = \pi(d_{pupil}/2)^2$ is the pupil area and $A_{retina} = 4\pi(d_{eye}/2)^2/2$ is the total area of the retina internally covering a half of the eye. Here we cannot then exclude a possible activation of rods in the mouse retina because they have similar absorption spectra to the M-opsin expressing cones (peak sensitivity at 498 and 508 nm, respectively) [9,32] and may escape from saturation even at $10^7$ R*/rod/s [33]. However, the rod system is likely optimized to work in the scotopic condition, and thus less affected by the

natural image statistics in the photopic condition. In the mouse retina, rods are indeed distributed more densely (~97% of all photoreceptors) and rather uniformly [34].

Given the average cone density $\rho_{cone}$ = 12,400 cells/mm$^2$ [34], the sampling resolution (or the "pixel size") of the mouse visual system is on the order of 0.25 degrees ($= 180/(\sqrt{\rho_{cone}} \cdot \pi d_{eye}/2)$ for photopic vision), and can go as high as 0.05 degrees if rod photoreceptors are also involved (average density, 437,000 cells/mm$^2$ [34]; or average diameter of 1.4 μm [35]). The spatial resolution of the acquired images (0.006 degrees/pixel) is thus good enough to cover the pixel size of the mouse vision.

## Image registration

The raw images from the two cameras (12 bit depth saved in the 16 bit grayscale Portable Network Graphic format, 1920-by-1200 pixels each) were pre-processed to form a registered image in Matlab (Image Processing Toolbox). First, we corrected the optical vignetting by normalizing the pixel intensity of the raw image $I_{raw}(x, y)$ for each channel by the ratio of the pixel and the maximum intensities of the reference image $I_{ref}(x, y)$:

$$I_{corrected}(x, y) = I_{raw}(x, y) \cdot \frac{\max[I_{ref}(x, y)]}{I_{ref}(x, y)}. \tag{1}$$

We next applied a two-dimensional median filter (3-by-3 pixel size) to remove salt-and-pepper noise from the corrected images for each channel. Then we applied a projective transformation based on manually selected control points to register the UV image to the green image. Finally, we manually cropped the two images to select only those areas in focus. The cropped images resulted in the pixel size ranging from 341 to 1766 pixels (2.0–10.6 degrees) in the horizontal axis and from 341 to 1120 pixels (2.0–6.7 degrees) in the vertical axes (see examples in Fig 2). We never changed the image resolution.

## Image analysis

We analyzed the first- and second-order image statistics of the obtained natural scenes in UV and green channels because the retina is not sensitive to higher-order statistics [36,37] (but see S4 Fig for higher-order statistics). Here we excluded a small set of the horizontal images ($N$ = 15) from the analysis, and focused on the following two major image groups: 1) looking-up images taken with a positive camera elevation angle ($N$ = 100), presumably falling in the ventral retina and thus perceived in the upper part of an animal's visual field; and 2) looking-down images with a negative camera elevation angle ($N$ = 117) perceived in the lower visual field (i.e., the dorsal retina). To ensure the separation between the image categories, we calculated the relative light intensity along the horizontal and vertical axes of each image category (S1 Fig). Specifically, we first corrected the pixel values of each image with the exposure length and the ND filter attenuation, and then normalized them by the mean pixel intensity value of all images. For the population analysis, the images were then aligned to the center in horizontal axes for all images, while to the top edge, center, or bottom edge in vertical axes for the lower, horizontal, upper visual field image categories, respectively. For each image data set, we used a sign-test to compare the image statistics parameter values between the UV and green channels (Figs 3–6; significance level, 0.05). All image analysis was done in Matlab (Mathworks).

**Light intensity normalization.** The visual system adapts its sensitivity to the range of light intensities in each environment [38,39]. We thus first normalized the pixel intensity of each UV and green image to have the intensity value ranging from zero to one (by subtracting the minimum value of the image, followed by the division by the maximum value), and then

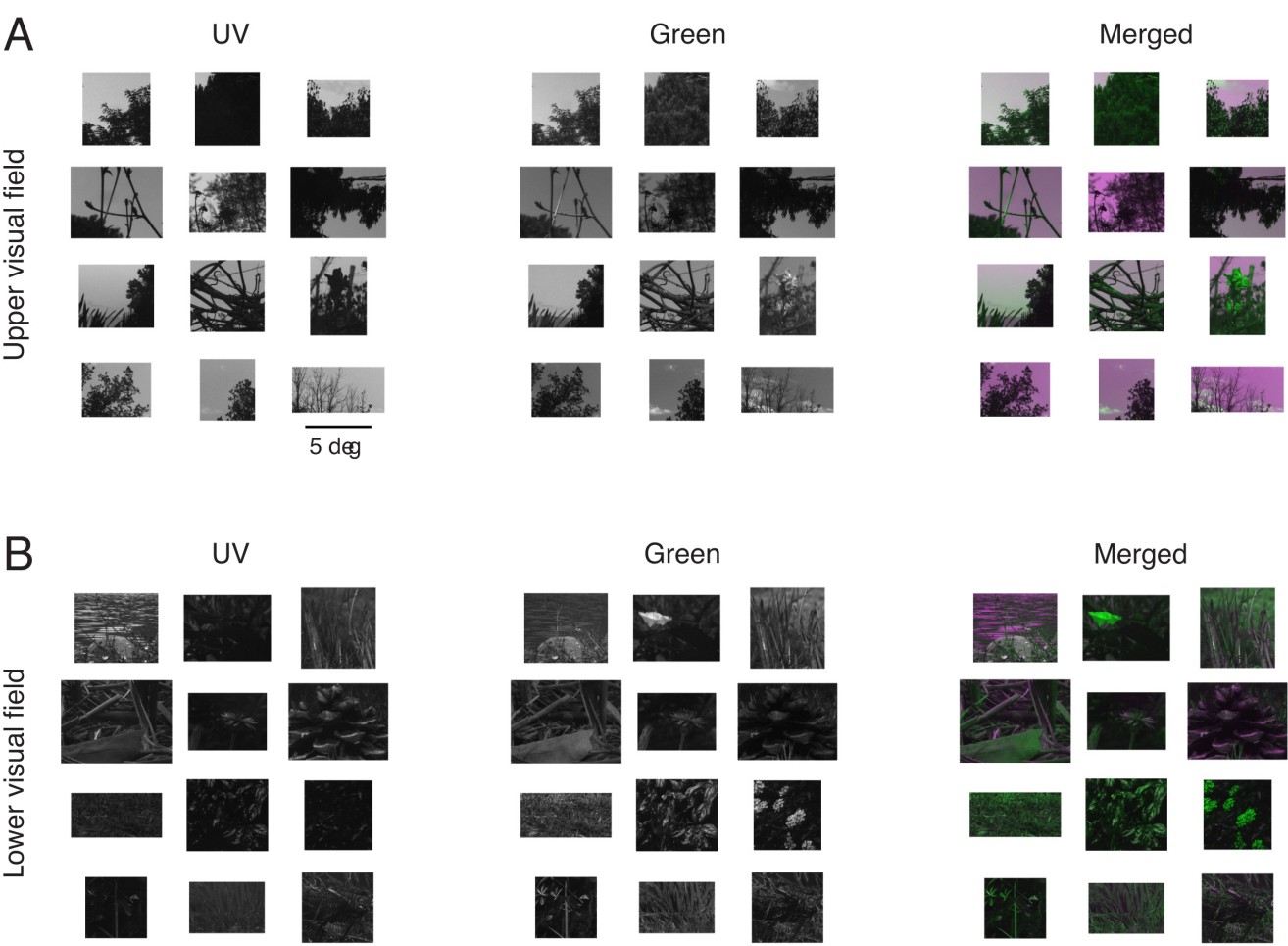

**Fig 2. Representative images of the natural scenes in UV and green channels.** See S2 Fig for the UV-Green pixel intensity distribution of these example images. (A) Upper visual field images taken with positive camera elevation angles (UV, Green, and pseudo-color merged images from left to right). These images typically contain trees and branches with sky backgrounds. (B) Lower visual field images taken with negative camera elevation angles, often containing a closer look of grasses and flowers.

calculated the histogram (bin size, 0.01) to compare the normalized intensity distributions of the UV and green images for the upper and lower visual fields (Fig 3A and 3B).

**Local contrast.** To calculate the local statistical structure of the normalized intensity images (Figs 3C, 3D and S3), we used the second-derivative (Laplacian) of a two-dimensional Gaussian filter:

$$\text{LoG}(x, y) = \frac{1}{\pi\sigma^4}\left(1 - \frac{x^2 + y^2}{2\sigma^2}\right)\exp\left[-\frac{x^2 + y^2}{2\sigma^2}\right], \tag{2}$$

with the standard deviation $\sigma$ = 5, 10, 20, 40 pixels for the spatial range $x, y \in [-3\sigma, 3\sigma]$. Here we chose a rather arbitrary size of the filter width (0.18–1.44 degrees) because natural image statistics are scale invariant (S3 Fig) [1,21]. The local contrast distribution was then fitted to the two-parameter Weibull distribution:

$$w(x) = \beta\gamma|x|^{\gamma-1}\exp[-\beta|x|^\gamma], \tag{3}$$

where $x$ is the local contrast value, $\beta > 0$ is the scale parameter (width) of the distribution, and

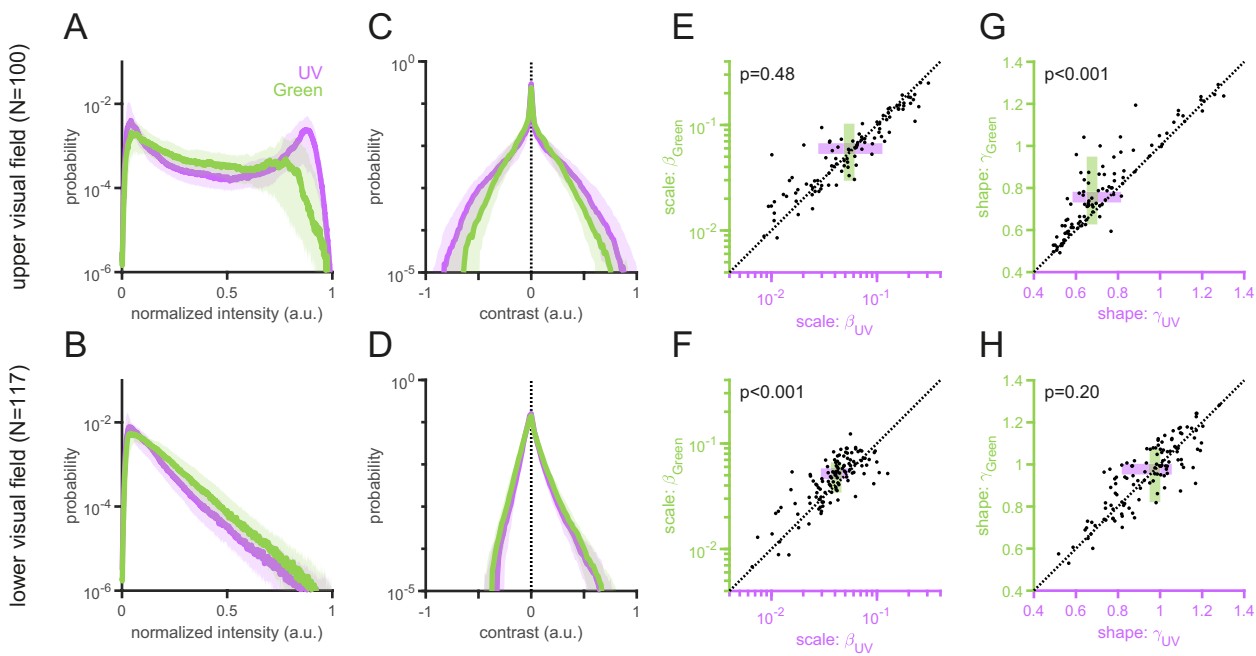

**Fig 3. Light intensity and local contrast distributions of the "mouse-view" natural images.** (A,B) Normalized light intensity distributions of the upper (A) and lower (B) visual field images for UV (violet) and Green (green) channels (median and interquartile range). (C,D) Local contrast distributions computed with the Laplacian-of-Gaussian filter ($\sigma = 10$ in Eq (2); see S3 Fig for the distributions computed with different $\sigma$ values). The distribution of the UV channel is more strongly heavy-tailed than that of the Green channel for the upper visual field images (C), but the Green channel's distribution is wider than the UV channel's for the lower visual field images (D). (E–H) Scale ($\beta$; E,F) and shape ($\gamma$; G,H) parameters from the Weibull distribution fitted to each image (Eq (3); see Methods for details). For the upper field images (E,G), the UV channel has significantly smaller $\gamma$ (G) but comparable $\beta$ (E) values than the Green channel. In contrast, for the lower field images (F,H), the Green channel has significantly larger $\beta$ (F) but comparable $\gamma$ (H) values than the UV channel. *P*-values are obtained from sign-tests.

$\gamma > 0$ is the shape parameter (peakedness). In particular, larger $\beta$ and smaller $\gamma$ values indicate wider and more heavy-tailed distributions, respectively, hence higher contrast in the images. Sign-tests were used to compare these parameter values between UV and green images (Fig 3E–3H).

**Achromatic and chromatic contrast.** To analyze the achromatic contrast of our image data sets (Fig 4), we calculated the root mean square (RMS) contrast $C_{RMS}^*(x, y)$ for each channel of normalized intensity images [22]:

$$C_{RMS}^*(x, y) = \frac{\sigma^*(x, y)}{\mu^*(x, y)}, \tag{4}$$

where $\mu^*(x, y)$ and $\sigma^*(x, y)$ are the mean and standard deviation of a circular image patch (radius, 30 pixels) centered at location $(x, y)$, respectively (S4 Fig, together with skewness and kurtosis as the third and fourth standardized moment, respectively, and entropy, $-\Sigma p \log p$, where $p$ is the probability distribution of the pixel intensity of the image patch); and the asterisk "*" is either "UV" or "Green" indicating the channel identity (Fig 4A and 4B). Chromatic contrast $C(x, y)$ was then defined as a difference of the RMS contrasts between the two channels (Fig 4C and 4D):

$$C(x, y) = C_{RMS}^{UV}(x, y) - C_{RMS}^{Green}(x, y). \tag{5}$$

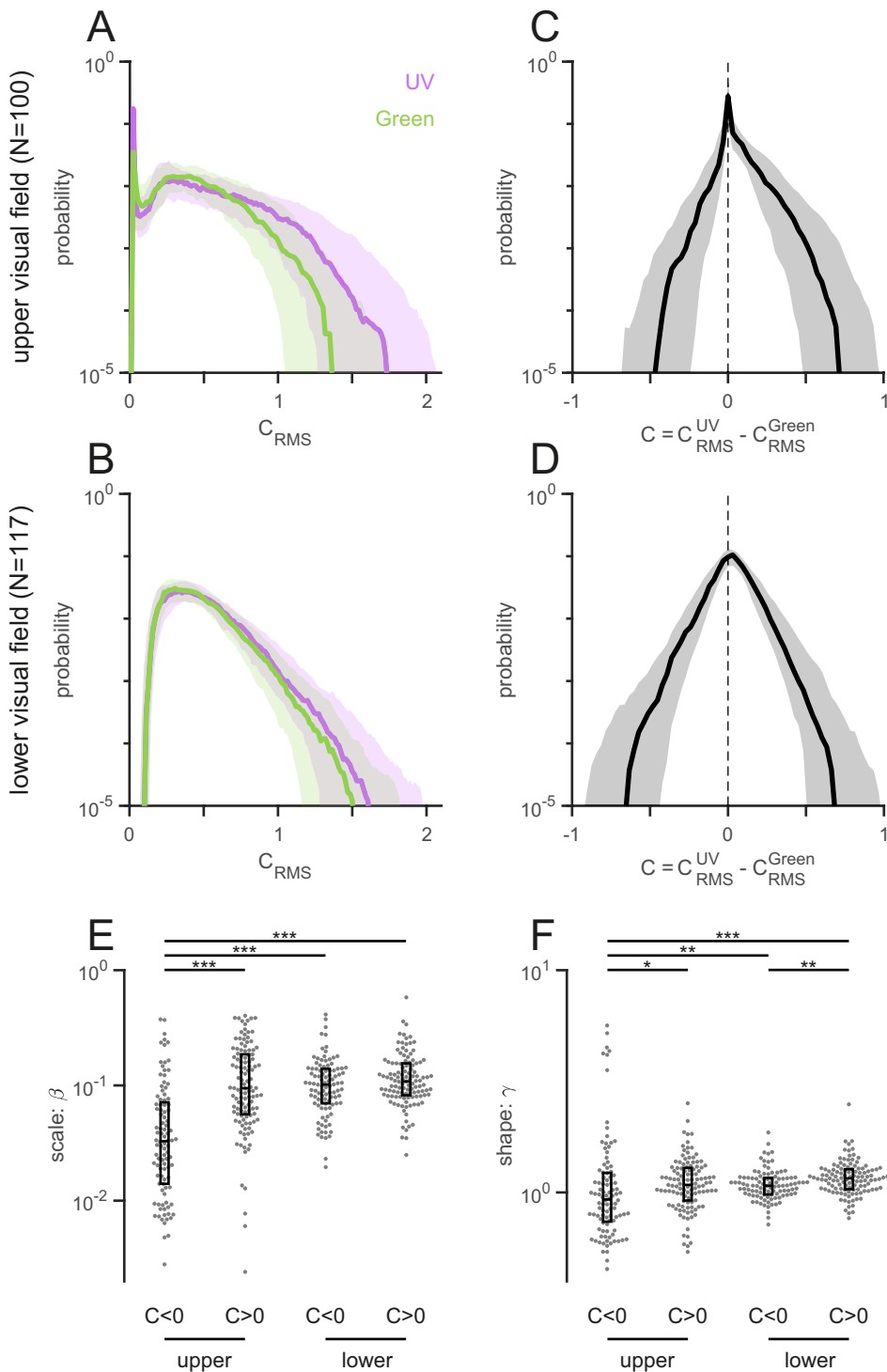

**Fig 4. Achromatic and chromatic contrast of "mouse-view" images.** (A,B) Root mean square (RMS) contrast of the upper (A) and lower (B) field images, computed independently for the UV (violet) and Green (green) channels of each image (local patch size, 30 pixel radius; Eq (4) in Methods). The UV channel has higher achromatic contrast, especially for the upper visual field images (median ± interquartile range). (C,D) Chromatic contrast distributions (median ± interquartile range) computed as a difference of the RMS contrasts between the UV and Green channels (Eq (5) in Methods). The distribution was asymmetric for the upper field images (C) but rather symmetric for the lower field images (D). (E,F): Scale (β; E) and shape (γ; F) parameters from the Weibull distribution fitted to each side

of the chromatic contrast distribution of each image. The box plot shows the median ± interquartile range. The upper field images contain fewer pixels that have higher contrast in Green than in UV (rank-sum test: Three stars "★★★" indicating $p<0.001$; ★★, $p<0.01$; and ★, $p<0.05$).

For quantification, we fitted the Weibull distribution (Eq (3)) to the left (C<0) and right (C>0) sides of the chromatic contrast distributions separately (Fig 4E and 4F).

**Power spectral density.** The power spectral density of the normalized intensity image $I(x, y)$ was computed with the fast Fourier transform (FFT; Fig 5):

$$F(\omega_x, \omega_y) = \text{FFT}[I(x, y)], \tag{6}$$

$$S(\omega_x, \omega_y) = F(\omega_x, \omega_y)F^*(\omega_x, \omega_y), \tag{7}$$

where the superscript $^*$ denotes complex conjugate, and $\omega_x$ and $\omega_y$ represent the horizontal and vertical spacial frequency (ranging from -0.5 to 0.5 cycles/pixel), respectively. As the average power spectrum of natural images generally falls with a form $1/f^\alpha$ over the spatial frequency $f$ with a slope $\alpha \sim 2$ [1,40,41], we fitted the power function $b/\omega^\alpha$ to $S(\omega_x, 0)$ and $S(0, \omega_y)$, where $a$ and $b$ indicate the slope and Y-intercept in the log-log space. We used a sign-test to compare these parameter values between UV and green channels (Fig 5I–5P).

**Spatial autocorrelation.** Following the Wiener–Khinchin theorem, the spatial autocorrelation $R(x, y)$ was computed with the inverse FFT of $S(\omega_x, \omega_y)$ in Eq (7):

$$R(x, y) = \text{IFFT}[S(\omega_x, \omega_y)], \tag{8}$$

where $x$ and $y$ represent horizontal and vertical distances of the two pixel points in the target image, respectively (Fig 6). Sign-tests were used to compare the $R(d_h, d_v)$ values at representative data points: $[d_h, d_v] = [0,50], [50, 0]$ (Fig 6I–6L).

## Results

### Multi-spectral camera for the mouse vision

The mouse retina expresses short (S)- and middle (M)-wavelength sensitive opsins that are maximally sensitive to ultraviolet (UV; $\sim 360$ nm) and green ($\sim 508$ nm) wavelengths of light, respectively [9–11]. Existing public databases of natural scenes contain a diverse set of images including both natural and artificial objects in both gray and color scales visible to humans [e.g., 42–45], but only a handful cover UV images [22–24]. To examine the natural image statistics of the mouse vision, especially for those of the upper and lower visual fields to test the optimality of the dorsoventral functional division of the mouse retina [9,10,18–20], we set out to build a multi-spectral camera system for acquiring images of the same scenes in both UV and green spectral domains (Fig 1).

We first modelled the spectral sensitivity of the mouse dichromatic vision to determine the center wavelengths of the two channels. Because the lens and cornea absorb shorter wavelength light (e.g., UV rays) more than longer wavelength light, we corrected the absorption spectra of the mouse cone photoreceptors [31] with the transmission spectra of the whole eye optics [30]. This resulted in a slight shift of the center wavelengths to a longer wavelength by several nanometers: from $\sim 360$ nm to $\sim 365$ nm for the S-cone and from $\sim 508$ nm to $\sim 512$ nm for the M-cone (Fig 1B). Thus, the ocular transmittance had only minor effects on the spectral sensitivity of the mouse vision, reassuring its sensitivity to near-UV light [20,46].

We then designed a multi-spectral camera system accordingly using a beam-splitting strategy (Fig 1A; see Methods for specifications) [25,26]. By convolving the measured transmission

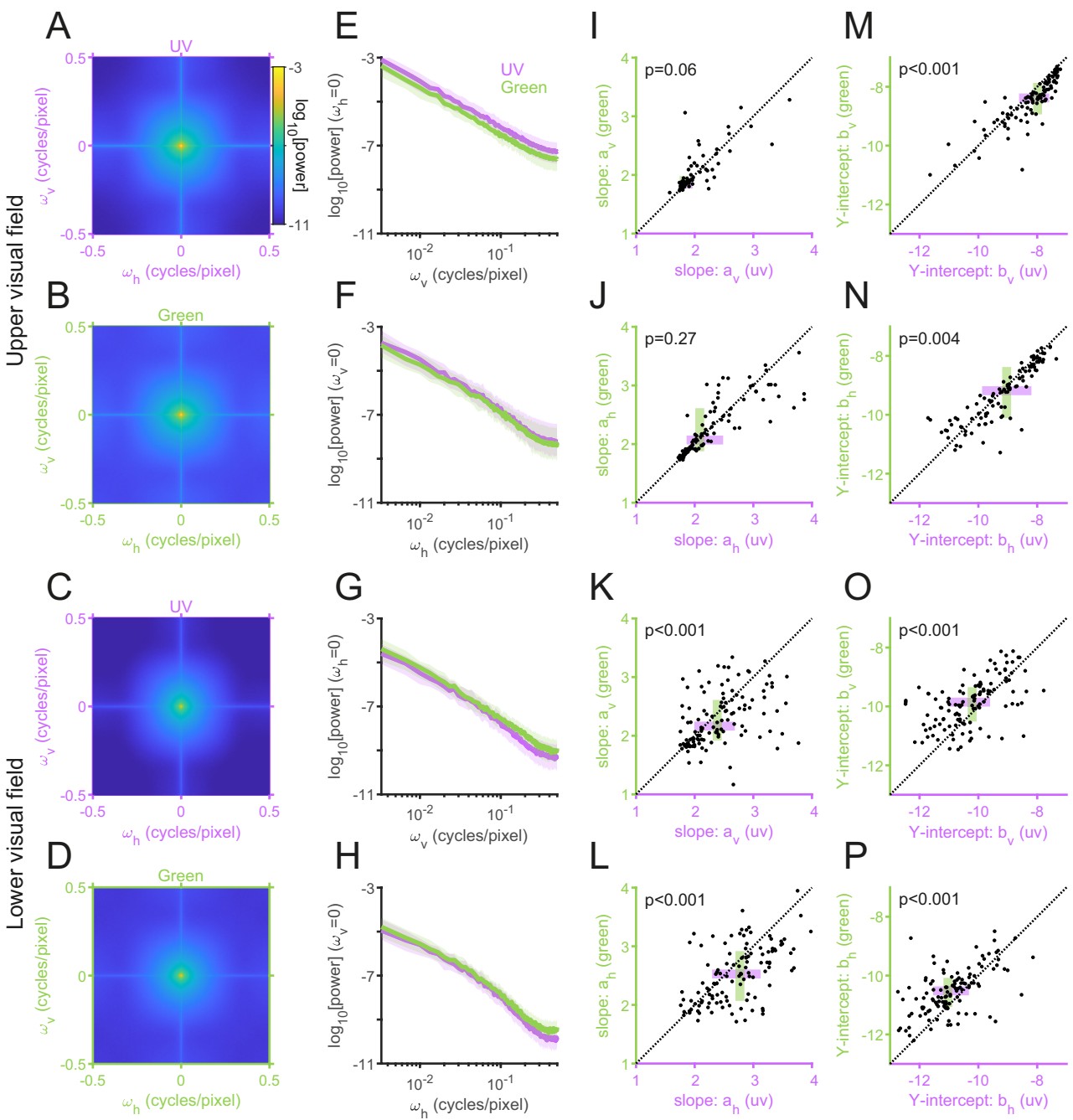

**Fig 5. Power spectrum of the "mouse-view" natural images.** (A–D) The average power spectra of the upper (A,B) and lower (C,D) visual field images for the UV (A,C) and Green (B,D) channels. (E–H) The power spectra in the vertical (E,G) and horizontal (F,H) directions (median and interquartile range) for the upper (E,F) and lower (G,H) visual field images. (I–M) The slope ($a$; I–L) and Y-intercept ($b$; M–P) parameters of the power function $b/\omega^a$ in the log-log space fitted to the power spectra of each image in the vertical (I,K,M,O) and horizontal (J,L,N,P) directions. For the upper visual field images (I,J,M,N), the UV channel has significantly larger $b$ (M,N) but comparable $a$ (I,J) values than the Green channel. For the lower field images (K,L,O,P), in contrast, the Green channel has significantly larger $b$ (O,P) and smaller $a$ (K,L) values than the UV channel. *P*-values are obtained from sign-tests.

spectrum of the camera optics with the sensitivity spectrum of the camera sensors [29], we identified that our imaging device had the sensitivities to ~368±10 nm and ~500±30 nm (center wavelength ± half-width at half maximum; HWHM) for the UV and green channels,

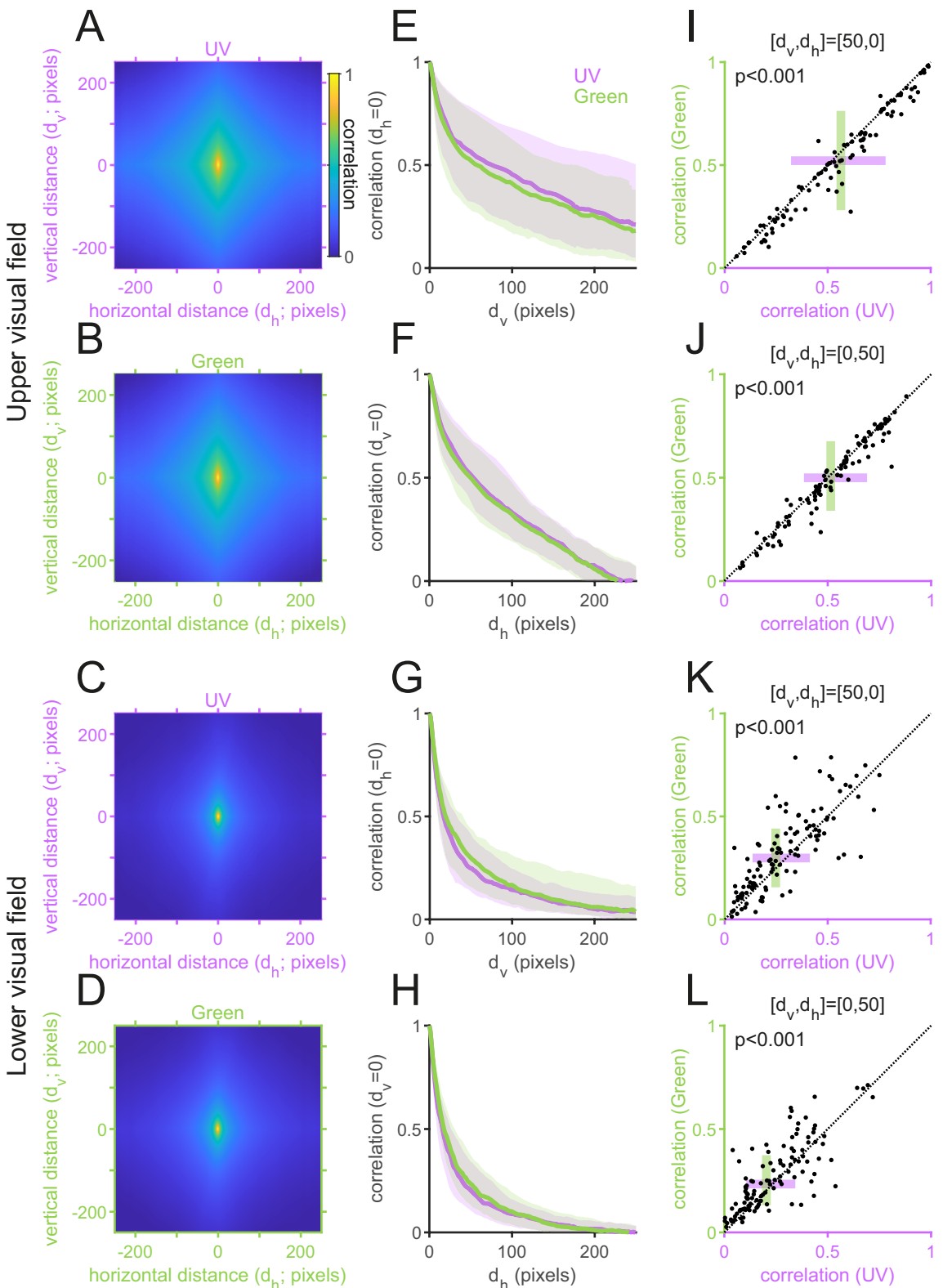

**Fig 6. Spatial autocorrelation of the "mouse-view" natural images.** (A–D) The average spatial autocorrelation of the upper (A,B) and lower (C,D) visual field images for the UV (A,C) and Green (B,D) channels, respectively. (E–H) The spatial autocorrelation in the vertical (E,G) and horizontal (F,H) directions (median and interquartile range). The UV channel has a higher and wider spatial correlation for the upper visual

field images (E,F), while the Green channel has a higher and wider spatial correlation for the lower visual field images (G,H). (I–L) Representative spatial correlation values of the pixels horizontally (I,K) or vertically (J,L) separated by 50 pixels for the upper (I,J) and lower (K,L) visual field images. *P*-values were obtained from sign-tests.

respectively (Fig 1B). This confirms that the UV and green channels of our device were spectrally well isolated, and that the two channels largely matched to the spectral sensitivity of the mouse vision [9–11].

## Ultraviolet and green image collection

To collect images that mice would encounter in their natural habitats, we went out to natural fields and wild forests in the countryside and mountain area of Lazio/Abruzzo regions in Italy across different seasons. We placed the multi-spectral camera on the ground at about a height of the mouse eye, and acquired images of natural objects alone at various distances (e.g., clouds, trees, flowers, and animals), excluding any artificial objects. These images were taken with different camera angles in the presence of ample natural light (S1 Fig). The images were preprocessed to correct optical vignetting and remove salt-and-pepper noise, and cropped to exclude areas out of focus on the edges (see Methods for details). This led to a set of 232 pairs of UV and green images of various "mouse-view" natural scenes.

Besides well-known facts that UV light is reflected well by open water and some plants [13,14], we noticed several distinct features between the UV and green images (see examples in Fig 2). First, clouds often appeared dark and faint in the UV images than in the green ones. In some cases, even negative contrast was formed for the clouds in UV while positive contrast in green. Second, fine textures were more visible in the green images than in the UV ones. In particular, objects in the upper field UV images were often dark in a nearly uniform manner due to back-light, whereas fine details of the objects were nevertheless visible in the corresponding green images despite a high contrast against the sky. For the lower field images, in contrast, distinct brighter spots stood out in UV due to reflections of shiny leaves and cortices, while more shades and shadows were visible in green. These qualitative observations already suggest that the UV and green images have distinct statistical properties.

## Normalized intensity and contrast distributions of UV and green images

To analyze the image statistics more formally, we first calculated the normalized intensity distribution of the UV and green channels for the upper and lower visual field images (Fig 3A and 3B). Because the visual system adapts its sensitivity to the range of light intensities in each environment [38,39], we normalized the pixel intensity of each UV and green image to be within the range from zero to unity. We then found that, for the upper visual field images, the probability distributions of both UV and green intensity values were bimodal (Fig 3A). The two peaks of the UV intensity distribution, however, were higher and more separated than those of the green intensity distribution, suggesting that luminance contrast is higher in UV than in green when animals look up. In contrast, the normalized intensity distributions of the lower field images were unimodal and skewed to the right for both color channels. The distribution was more strongly heavy-tailed for the green than for the UV images (Fig 3B), indicating higher contrast in green than in UV when animals look down.

To better examine the contrast in the two different spectral domains, we calculated the local image contrast using the second derivative (Laplacian) of a two-dimensional Gaussian filter (Eq (2) in Methods). This filter follows the antagonistic center-surround receptive fields of early visual neurons (e.g., retinal ganglion cells [47,48]) that are sensitive to local contrast, and is commonly used for edge detection in computer vision [49–51]. Consistent with what was

implicated by the intensity distributions (Fig 3A and 3B), we found that 1) the probability distribution of local contrast was generally wider for the upper visual field images than for the lower visual field images; and 2) the local contrast distribution was wider for the upper visual field UV images than for the corresponding green images (Figs 3C, S3A, S3C and S3E), but narrower for the lower visual field UV images than for the green counterparts (Figs 3D, S3B, S3D and S3F). To quantify these differences, we fitted a two-parameter Weibull function (Eq (3) in Methods) to the local contrast distribution of each image in each channel [52,53], where the first scale parameter ($\beta$) describes the width of the distribution, hence a larger value indicating higher contrast; and the second shape parameter ($\gamma$) relates to the peakedness, with a smaller value indicating a heavier tail and thus higher contrast in the image. For the images above the horizon, the UV channel had significantly smaller shape parameter values than the green channel (Fig 3G) with comparable scale parameter values (Fig 3E). In contrast, for the images below the horizon, the green channel had significantly larger scale parameter values than the UV channel (Fig 3F), with no difference in the shape parameter values (Fig 3H). Thus, the image statistics showed distinct characteristics between the upper and lower visual field image data sets, with higher contrast in UV than in green for the upper visual field images, and vice versa for the lower visual field images.

Importantly, such differences in the local contrast distributions do not agree well with what the efficient coding hypothesis implies from the physiological and anatomical properties of the mouse retina [3,4]. Solely from an information theoretic viewpoint, a narrower contrast distribution is better encoded with a more sensitive cone type to maximize its bandwidth [54]. In the mouse retina, the functional S-cones are more sensitive to contrast than the functional M-cones [17–20,46]; and the functional S-cones are denser towards the ventral part of the retina, preferentially sampling the upper part of the visual field, while the functional M-cones towards the dorsal retina, sampling the lower visual field [15,16,18]. Therefore, this particular retinal organization is optimal if the upper visual field images had lower contrast in UV than in green, and the lower visual field images had higher contrast in UV than in green. Our image analysis, however, showed the opposite trend in the "mouse-view" visual scenes (Fig 3).

## Achromatic and chromatic contrast of "mouse-view" images

To examine achromatic and chromatic contrast of our image data sets, we next measured the root mean square (RMS) contrast (Eqs (4) and (5) in Methods) that is commonly used in psychophysical studies [22]. We found that the achromatic RMS contrast (Eq (4)) was higher in UV than in green channels, especially for the upper visual field images (Fig 4A and 4B). The upper visual field images then had an asymmetric chromatic contrast distribution (Eq (5); Fig 4C), where pixels with higher contrast in UV than in green were more abundant than those with higher contrast in green than in UV (Fig 4E and 4F). In contrast, the chromatic contrast distribution was rather symmetric for the lower visual field images (Fig 4D), and it was overall wider than that for the upper visual field images (Fig 4E and 4F).

This indicates that UV-green chromatic information exists across the visual field, even though the exact shape of the chromatic contrast distribution may depend on the image contents [22]. We indeed identified UV-green chromatic objects in both lower and upper visual field images (see examples in Figs 2 and S2) and thus cannot explain why the mouse retina has chromatic circuitry preferentially on the ventral side (upper visual field) [55–57]. In principle, mice could retrieve UV-green chromatic information across the visual field, given that 1) genuine S-cones and rods are distributed rather uniformly across the mouse retina [34]; 2) rods have similar absorption spectra to M-cones (peak sensitivity at 498 and 508 nm, respectively; Fig 1B) [9,32]; and 3) rods can escape from saturation even under photopic conditions [33].

Larger image datasets sampled under more diverse conditions are required to assess the optimality of the chromatic circuitry in the mouse retina, especially because the rod system plays a role not only in the color vision but also in the scotopic vision.

### Power spectrum and autocorrelation of UV and green images

We next analyzed the second-order statistics of the acquired images. Specifically, we computed the power spectrum (Fig 5) and spatial autocorrelation that describes the relationship of the two pixel intensity values as a function of their relative locations in the images (Fig 6; see Methods for details). As expected [1,21], the power spectra generally followed $1/\omega^a$ on the spatial frequency $\omega$ for both UV and green channels irrespective of the camera angles (in log-log axes; Fig 5A–5H); and were higher for the vertical direction than for the horizontal direction (Fig 5A–5H)—i.e., the spatial autocorrelation was elongated in the vertical direction (Fig 6A–6D).

There are, however, several distinct properties between the UV and green channels for the upper and lower visual field images. First, the slope of the power spectra $a$ was larger for the lower visual field images than for the upper visual field images (Fig 5I–5L); equivalently, the spatial autocorrelation was narrower for the lower visual field images (Fig 6E–6H), indicating the presence of more fine textures in those images. Second, for the upper visual field images, the UV power spectra were higher than the green ones in both vertical and horizontal directions (e.g., the Y-intercept $b$, indicating the log-power at the spatial frequency of 1 cycle/pixel; Fig 5M and 5N). In contrast, for the lower visual field images, the UV power spectra were lower with a larger slope than the green counterparts (Fig 5K, 5L, 5O and 5P). Equivalently, the spatial autocorrelation was wider in UV than in green for the upper visual field images, and vice versa for the lower visual field images (Fig 6E–6H).

Under an efficient coding hypothesis, a higher spatial autocorrelation implies that less cones are needed to faithfully encode the scenes [3,4,54]. One would then expect from the "mouse-view" image statistics that the functional S- and M-cones should be denser on the dorsal and ventral parts of the mouse retina, respectively, to achieve an optimal sampling. However, the opposite is the case with the mouse retina [15,16,18], suggesting that the cone distribution bias in the mouse retina cannot be simply explained by the optimality principle from an information theoretic viewpoint.

## Discussion

To study the natural image statistics for the mouse vision, here we collected a set of 232 "mouse-view" two-color images of various natural scenes across different seasons using a custom-made multi-spectral camera (Figs 1 and 2). We identified distinct image statistics properties for the two channels between the images above and below the horizon (Figs 3–6 and S4). Specifically, both the local contrast and the spatial autocorrelation were higher in UV than in green for the upper visual field images, while they were both lower in UV than in green for the lower visual field images. This disagrees with what the efficient coding hypothesis implies [3,4] from the functional division of the mouse retina along the dorsoventral axis [15,16,18]. We thus suggest that the given retinal organization in mice should have evolved not only to efficiently encode natural scenes from an information theoretic perspective, but likely to meet some other ethological demands in their specific visual environments [22].

How faithful are our images to what mice actually see in their natural habitats? This is a critical question because image statistics depend on the quality and contents of the images. Our camera system was designed to collect high-quality UV-green images (Figs 1 and 2) comparable to the existing natural image datasets for human vision [42–45]. However, caveats include that 1) the effects of the mouse eye optics were not considered in the image acquisition or

analysis; 2) no motion dynamics were considered; 3) images were taken under ample light during the day, while mice are nocturnal; and 4) our image datasets were still relatively small and did not cover the entire visual field for the mouse vision. It is a future challenge to address these questions, for example, by measuring the properties of the mouse eye optics, simulating images projected onto the mouse retina, and analyzing the statistics of these images.

### "Mouse-view" natural image database

We employed a beam-splitting strategy to simultaneously acquire UV and green images of the same scenes (Fig 1) because it has certain advantages over other hyper- or multi-spectral imaging techniques [25,26]. First, a previous study used a hyperspectral scanning technique where a full spectrum of each point in space was measured by a spectrometer [18]. While the photoreceptor response could be better estimated by using its absorption spectra, the scanned images through a pinhole aperture inevitably had lower spatial and temporal resolutions than the snapshot images acquired with our device. Second, a camera array can be used for multispectral imaging with each camera equipped with appropriate filters and lenses [58]. This is easy to implement and will perform well for distant objects; however, because angular disparity becomes larger for objects at a shorter distance, one would have a difficulty in taking close-up images that small animals such as mice would normally encounter in their everyday lives. Finally, our single-lens-two-camera design is simple and cost-effective compared to other snapshot spectral imaging methods [26]. In particular, commercially available devices are often expensive and inflexible, hence not suitable for our application to collect images that spectrally match the mouse vision.

There are several conceivable directions to expand the "mouse-view" natural image database. First, we could take high dynamic range images using a series of different exposure times. This works only for static objects, but can be useful to collect images at night during which nocturnal animals such as mice are most active. Second, we could take a movie to analyze the space-time statistics of natural scenes [22]. It would be interesting to miniaturize the device and mount it on an animal's head to collect time-lapse images with more natural self-motion dynamics [59,60]. Expanding our "mouse-view" natural image datasets will be critical to better understand the visual environment of mice and develop a theoretical explanation on species-specific and non-specific properties of the mouse visual system.

### Optimality of the mouse retina

What selective pressures have driven the mouse retina to favor UV sensitivity over blue and evolve the dorsoventral gradient in the opsin expression? Our image analysis suggests that the coding efficiency alone with respect to the natural image statistics cannot fully explain the distinctive organization of the mouse retina (Figs 3–6). For example, we argued from an information theoretic viewpoint that, for equalizing the bandwidth within the system, high contrast images in the upper visual field (Fig 3C) should be encoded with less sensitive photoreceptors (M-cones), while low contrast images in the lower visual field (Fig 3D) with more sensitive photoreceptors (S-cones) [18]. In contrast, one could also argue from an ethological viewpoint that more sensitive S-cones are driven more strongly by high contrast images in the upper visual field and thus better suited to process biologically relevant information, such as aerial predators [2,22].

To understand in what sense the mouse retina's organizations are optimal, one then needs to clarify visual ethological demands that are directly relevant for survival and reproduction. For example, fresh mouse urine reflects UV very well, and this has been suggested to serve as a con-specific visual cue for their territories and trails besides an olfactory cue [61]. The UV

sensitivity can also be advantageous for the hunting behavior of mice because many nocturnal insects are attracted to UV light. Furthermore, increased UV sensitivity in the ventral retina may improve the detection of tiny dark spots in the sky, such as aerial predators [62]. Indeed, the S-opsin-dominant cones in mice have higher sensitivity to dark contrasts than the M-opsin-dominant ones [18], and turning the anatomical M-cones into the functional S-cone by co-expressing the S-opsin will dramatically increase the spatial resolution in the UV channel because the mouse retina has only a small fraction of the uniformly distributed genuine S-cones (∼5%) compared to the co-expressing cones (∼95%) [11,16,17,63].

These arguments, however, are difficult to generalize because each species has presumably taken its own strategy to increase the fitness in its natural habitat, leading to convergent and divergent evolution. On the one hand, UV sensitivity was identified in some mammals that live in a different visual environment than mice, including diurnal small animals such as the degu and gerbil [61,64,65] and even large animals such as the Arctic reindeer [66]. On the other hand, some species showing a similar behavioral pattern as mice do not have the dorso-ventral division of the retinal function [12–14]. For example, even within the genus *Mus*, some species do not have the dorsoventral gradient of the S-opsin expression, and others completely lack the S-cones [67]. It is even possible that the cone distribution bias may have nothing to do with the perception of the color vision, but may arise just because of the developmental processes. Indeed, the center of the human fovea is generally devoid of S-cones [68,69], and there is a huge diversity in the ratio of M- and L-cones in the human retina across subjects with normal color vision [70,71]. Behavioral tests across species will then be critical for validating the ethological arguments to better understand the structure and function of the visual system [2]. We expect that the "mouse-view" natural image datasets will contribute to designing such studies.

## Supporting information

**S1 Fig. Relative pixel intensities along horizontal and vertical axes.** Relative pixel intensities (median ± interquartile range; UV and green channels in violet and green, respectively) were computed along horizontal (A,C,E) and vertical (B,D,F) axes for three different image categories based on the camera angle: Lower (A,B; N = 117), horizontal (C,D; N = 15), and upper (E, F; N = 100) visual field images. Pixel intensity did not change much horizontally but was generally lower in the lower field images (A,B) than in the upper field images (E,F). Discontinuity between the top edge of the lower field images (B, x-axis value of 0) and the bottom edge of the upper field images (F, x-axis value of 0) supports a good separation of the two image categories.
(PDF)

**S2 Fig. UV-Green pixel intensity distributions of representative "mouse-view" images.** Each scatter plot shows the distribution of the UV-Green pixel values from the corresponding image shown in Fig 2 (A, upper visual field images; B, lower visual field images). Virtually all pixels were within the dynamic range of the camera sensor (Sony, IMX174 CMOS; 12-bit depth saved in a 16-bit format).
(PDF)

**S3 Fig. Local contrast distributions of the natural scenes are scale invariant.** Local contrast distributions computed with different Laplacian-of-Gaussian filter sizes (A,B, $\sigma$ = 5; C,D, $\sigma$ = 20; E,F, $\sigma$ = 40; Eq (2)) are shown in the same format as Fig 3C and 3D ($\sigma$ = 10). The upper visual field images (A,C,D) generally showed higher contrast than the lower visual field images (B,D,F), especially for the UV channel (violet). The filter size (0.18–1.44 degrees) used in this

study is smaller than the receptive field size of mouse retinal ganglion cells (3–13 degrees) [72,73]. Given the scale invariance [2,21], however, we expect that our analysis results should hold for larger filters as well [22].
(PDF)

**S4 Fig. Natural image statistics for "mouse-view" images have distinct spectral properties between upper and lower visual fields across different order statistics.** The first- to the fourth-order image statistics (mean, A, B; standard deviation, C, D; skewness, E, F; kurtosis, G, H) as well as entropy (I, J) were computed for local images patches (0.36 degrees; UV, violet; Green, green). Joint (top) and marginal (bottom) probability distributions were then generated for the upper (A, C, E, G, I) and lower (B, D, F, H, J) visual field images.
(PDF)

## Acknowledgments

We thank all the members of the Asari lab at EMBL Rome for many useful discussions. EMBL IT Support is acknowledged for provision of computer and data storage servers.

## Author Contributions

**Conceptualization:** Hiroki Asari.

**Formal analysis:** Hiroki Asari.

**Investigation:** Luca Abballe.

**Resources:** Luca Abballe.

**Software:** Luca Abballe.

**Writing – original draft:** Hiroki Asari.

**Writing – review & editing:** Hiroki Asari.

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
