## [Decision Letter · Decision Letter 0]

22 Jun 2021

PONE-D-21-16635

Natural Image Statistics for Mouse Vision

PLOS ONE

Dear Dr. Asari,

Thank you for submitting your manuscript to PLOS ONE. After careful consideration, we feel that it has merit but does not fully meet PLOS ONE’s publication criteria as it currently stands. Therefore, we invite you to submit a revised version of the manuscript that addresses the points raised during the review process.

Please address all reviewers' comments in your response.

We look forward to receiving your revised manuscript.

Kind regards,

Manuel Spitschan

Academic Editor

PLOS ONE

Journal Requirements:

Reviewers' comments:

Reviewer's Responses to Questions

**Comments to the Author**

1. Is the manuscript technically sound, and do the data support the conclusions?

Reviewer #1: Yes

Reviewer #2: Partly

2. Has the statistical analysis been performed appropriately and rigorously? 

Reviewer #1: Yes

Reviewer #2: Yes

3. Have the authors made all data underlying the findings in their manuscript fully available?

Reviewer #1: Yes

Reviewer #2: Yes

4. Is the manuscript presented in an intelligible fashion and written in standard English?

Reviewer #1: Yes

Reviewer #2: Yes

5. Review Comments to the Author

Reviewer #1: The article by Abballe and Asari entitled " Natural Image Statistics for Mouse Vision" demonstrated a camera system for taking photos with dichromatic spectra: UV and green, corresponding to sensitivities of S-opsin (UV) and M-opsin (green) for the cone photoreceptors of mouse retina. Using this camera system, they acquired photos of natural scenes without artificial objects. The authors divided the image data into two main groups (upper and lower vision field). By analyzing 1) the pixel-by-pixel intensity (first order statistics) and 2) luminance contrast, power spectrum and spatial autocorrelation (second order statistics) in the two chromatic channels, the authors found distinct properties between two channels: higher local contrast and spatial autocorrelation in UV in the upper visual field, while they were lower in the lower visual field.

Comments/questions/suggestions:

1) This camera system used a 50mm lens to capture the nature sense. It would be good to know what the field of view (FOV) of the whole camera system is. Did they ever change the image resolution? In the mouse retina, the spatial resolution is determined by the receptive field (RF) size of the ganglion cells. The visual angle of different ganglion cell types varies from 3 to 13 degree. Thus, the local contrast may change with the RF sizes. Since the authors calculated the local contrast using a fixed area XY in [-30 pixels, +30 pixels], the local contrast may only represent the contrast for certain ganglion cell types.

2) In Figure3 A&B, the authors plotted the normalized intensity across the data sets. I am wondering if the authors did the calibration for the camera system. The camera may be underexposure for dark scenes and overexposure for bright scenes, which depends on camera settings. Both underexposure and overexposure can result in missing luminance information. There are quite pixels with 0 value in the figure, indicating that images tended to be underexposure. However, these data are normalized between 0-1. It is difficult to check their original values. Moreover, the authors used two different ND filters in green channel to increasing the dynamic range. In my view, this design also changed the sensitive of the green part even though the camera settings remained same. So, two images taken with different ND filters cannot be compared for contrast. Thus, the promising solution is to calibrate the system, relating the intensity of each pixel to the power of the light at defined wavelength. Another advantage of calibration can avoid gamma correction of any graphic-related system.

3) In the abstract, “… natural image statistics of UV light have never been extensively assessed beyond spectral analysis. …”. However, in the discussion part (line: 339), the authors mentioned a very similar camera system with related statistics (now published as Qiu et al., 2021). Could the authors compare the difference between these two camera systems?

4) In the multi-spectral camera design part (line 84), this dichroic filter (Edmund Optics, 68-330) is just a silver mirror according to Edmund Optics. Could the authors double check the information of this item?

5) Spatial autocorrelation is used to measure spatial dependence/similarity. In this study, this method calculated the autocorrelation within individual images, which depends on the content of each image. My hypothesis is that the results might be affected by the exposure settings for each scene in different chromatic channels. For example, right exposure of a tree can give more details and, hence, lower values, while underexposure can have higher correlation due to less details. Could the authors provide some data for different exposure settings from a same nature scene?

Reviewer #2: Abballe and Asari describe a UV/Green camera system that they have used to capture images of natural scenes in two spectral bands (a multi-spectral camera). They in turn analyse the natural image statistics in these two channels, and in two broad regions of the visual field (upper and lower). They find that the statistics differ across the spectral bands and regions, such that local contrast and spatial correlation are higher in the UV images taken in the upper visual field (compared with green), and lower in the UV images taken in the lower visual field (compared with green). In turn, they conclude that the mouse retina is not optimised to detect these differences, since the efficient coding hypothesis would predict optimal sampling with UV cones in the dorsal retina and green cones in the ventral (opposite to the known distribution). This work is novel at the time of submission, and will be valuable to the large community of researchers working on mouse visual physiology, in particular, a growing field researching the relevance of natural visual stimuli in visual coding.

I have a few questions that the authors should address:

I’d firstly like to know more about the camera and its calibration. In particular: what is its resolution (8 bit?); how linear is the detector, does it apply a gamma correction and if so, is it accounted for? Similarly, how sensitive is the camera to shorter (<400nm) wavelengths, and does this need to be accounted for in any way for the UV measurements?

Most of the analysis compares between image pixels – some frame of reference would be useful to understand how these pixels relate to the mouse visual system. What is the camera’s field of view in degrees, and what would be the size of each pixel be in terms of visual angle? How does that relate to the receptive field size of neurons in the mouse’s eye?

Relating to this, how ‘large’ are the images in approximate visual space, and what area of the retina would this likely cover? Are the images of upper and lower regions likely to occur within the same angle of view i.e. covering both dorsal and ventral parts of the retina, as the authors use for their analysis?

A similar analysis has been published while this paper has been under review (Qiu et al, 2021, Current Biology). Their data are in line with Abballe and Asari, though they conclude that the mouse retina’s specialisations match (rather than oppose) the scene statistics - some discussion is probably needed here.

Can the authors make any predictions about chromatic contrast, and how their measurements may relate to this aspect of vision? Relatedly, what about rod photoreceptors? There is increasing evidence that rods are also active in brighter light, and at least some discussion of this fact is needed in this manuscript.

6. PLOS authors have the option to publish the peer review history of their article (what does this mean?). If published, this will include your full peer review and any attached files.

Reviewer #1: No

Reviewer #2: No

---

## [Author Response · Author response to Decision Letter 0]

29 Jul 2021

We have addressed all the reviewers' concerns. Please see attached the response to the reviewers. We have also reformatted our manuscript to meet the PLOS ONE's style requirement.

---

## [Decision Letter · Decision Letter 1]

6 Oct 2021

PONE-D-21-16635R1Natural Image Statistics for Mouse VisionPLOS ONE

Dear Dr. Asari,

Thank you for submitting your manuscript to PLOS ONE. After careful consideration, we feel that it has merit but does not fully meet PLOS ONE’s publication criteria as it currently stands. Therefore, we invite you to submit a revised version of the manuscript that addresses the points raised during the review process.

Please address all comments raised by Reviewer #1.

We look forward to receiving your revised manuscript.

Kind regards,

Manuel Spitschan

Academic Editor

PLOS ONE

Journal Requirements:

Additional Editor Comments (if provided):

Reviewers' comments:

Reviewer's Responses to Questions

**Comments to the Author**

1. If the authors have adequately addressed your comments raised in a previous round of review and you feel that this manuscript is now acceptable for publication, you may indicate that here to bypass the “Comments to the Author” section, enter your conflict of interest statement in the “Confidential to Editor” section, and submit your "Accept" recommendation.

Reviewer #1: All comments have been addressed

Reviewer #2: All comments have been addressed

2. Is the manuscript technically sound, and do the data support the conclusions?

Reviewer #1: Yes

Reviewer #2: Yes

3. Has the statistical analysis been performed appropriately and rigorously? 

Reviewer #1: Yes

Reviewer #2: Yes

4. Have the authors made all data underlying the findings in their manuscript fully available?

Reviewer #1: Yes

Reviewer #2: Yes

5. Is the manuscript presented in an intelligible fashion and written in standard English?

Reviewer #1: Yes

Reviewer #2: Yes

6. Review Comments to the Author

Reviewer #1: The authors have made an extensive effort to address my concerns. The manuscript is improved a lot. I am afraid I remain unclear on the following points:

1) The title “Natural Image Statistics for Mouse Vision” does not specifically fit for what is described in the paper. The natural image statistics include many aspects such as first-order, second-order and higher-order statistics. In this study, only the contrast was massively analysed, while many other aspects remain untouched. In addition, the camera system only respected the spectra of the mouse retina. But the visual angle may only represent the FOV of no more than 5 RGCs. Although the authors expressed that “… natural image statistics are scale invariant…” (line 164-165) and I also agree with this in terms of statistics, this small FOV (< 11.3 degrees) cannot represent mouse vision (180 degrees).

2) The authors emphasized the advantages of this camera system by 1) smaller FOV for low distortion, 2) high-performance sensor with higher dynamic range and higher resolution, and 3) fewer optical elements ensure transmission of near UV (line 72-81). Nevertheless, this camera system was designed for mouse vision. So, the small FOV probably has the bad impression compared to real mouse vision (180 degrees). Meanwhile, Increasing the FOV can spontaneously solve the problem of focal depth. Also, the mouse retina has low resolution output depends on the density of RGCs (3 to 13 degrees). Thus, the described advantages of FOV (50mm lens) and high resolution (camera sensor) mostly refer to the human eye’s perspective. In addition, the optical elements are not fewer compared to the other system cited in the paper. There are already 6 lenses in 5 groups from the Nikon commercial camera lens and two filters in each chromatic channel.

3) As the authors did not perform the intensity calibration of the camera system, I would like to suggest doing baseline/background subtraction before the normalization. The high-performance sensor has a big dynamic range. However, it still has its minimum detectable signal, like other sensors in the market. This effect can be nicely explained from revised Fig. 1A inset. The line was not able to pass through the origin of the coordinate system and the y-intercept is the effective minimum sensitivity of the camera sensor. When reflecting in the images, it was exactly shown by the UV-green pixel intensity distributions (S2 Fig) where one cannot find any values at 0, instead the darkest points were accumulated at the values close to zero. In fact, those data dots contain no real luminance information, meaning underexposure, which should be excluded in the statistic calculations. In this case, contrast should be calculated based on images with baseline/background subtraction.

4) In the results part, the authors expressed no public database of natural scenes covering UV images (line 205-207). Here I would like to suggest the authors check the three publications below:

-Differt & Möller (2015) Insect models of illumination-invariant skyline extraction from UV and green channels. J. Theor. Biol. 380, 444–462

-Differt & Möller (2016) Spectral skyline separation: extended landmark databases and panoramic imaging. Sensors 16, 1614

-Qiu et al (2021) Natural environment statistics in the upper and lower visual field are reflected in mouse retinal specializations. Curr Biol. 2021 Aug 9;31(15):3233-3247.e6.

5) Again, there is a mistake of the hardware information that the short pass filter (Edmund Optics 84-708) should cut at 550nm, not 500nm.

Nevertheless, the method is appropriately described here, and the paper as a whole represents an important contribution to the public database for nature images of UV.

Reviewer #2: Thanks to the authors for responding to my comments and questions, I believe they have addressed all my points well.

7. PLOS authors have the option to publish the peer review history of their article (what does this mean?). If published, this will include your full peer review and any attached files.

Reviewer #1: No

Reviewer #2: No

---

## [Editor Report · Decision Letter 2]

5 Jan 2022

Natural Image Statistics for Mouse Vision

PONE-D-21-16635R2

Dear Dr. Asari,

We’re pleased to inform you that your manuscript has been judged scientifically suitable for publication and will be formally accepted for publication once it meets all outstanding technical requirements.

Kind regards,

Manuel Spitschan

Academic Editor

PLOS ONE
---

## [Editor Report · Acceptance letter]

10 Jan 2022

PONE-D-21-16635R2 

Natural Image Statistics for Mouse Vision 

Dear Dr. Asari:

I'm pleased to inform you that your manuscript has been deemed suitable for publication in PLOS ONE. Congratulations! Your manuscript is now with our production department. 

Kind regards, 

on behalf of

Dr. Manuel Spitschan 

Academic Editor

PLOS ONE